# Industry 4.0 Quantum Strategic Organizational Design Configurations. The Case of Two Qubits: One Reports to One

**DOI:** 10.3390/s20236977

**Published:** 2020-12-06

**Authors:** Javier Villalba-Diez, Rosa María Benito, Juan Carlos Losada

**Affiliations:** 1Hochschule Heilbronn, Fakultät Management und Vertrieb, Campus Schwäbisch Hall, 74523 Schwäbisch Hall, Germany; 2Complex Systems Group, Escuela Técnica Superior de Ingenieros Agrónomos, Universidad Politécnica de Madrid, Av. Puerta de Hierro 2, 28040 Madrid, Spain; rosamaria.benito@upm.es (R.M.B.); juancarlos.losada@upm.es (J.C.L.)

**Keywords:** quantum strategic organizational design, industry 4.0, quantum circuits

## Abstract

In this paper we investigate how the relationship with a subordinate who reports to him influences the alignment of an Industry 4.0 leader. We do this through the implementation of quantum circuits that represent decision networks. In fact, through the quantum simulation of strategic organizational design configurations (QSOD) through five hundred simulations of quantum circuits, we conclude that there is an influence of the subordinate on the leader that resembles that of a harmonic under-damped oscillator around the value of 50% probability of alignment for the leader. Likewise, we have observed a fractal behavior in this type of relationship, which seems to conjecture that there is an exchange of energy between the two agents that oscillates with greater or lesser amplitude depending on certain parameters of interdependence. Fractality in this QSOD context allows for a quantification of these complex dynamics and its pervasive effect offers robustness and resilience to the two-*qubit* interaction.

## 1. Introduction

This work is designed as a brief disclosure of a significant new application of previous work on Quantum Strategic Organizational Design (QSOD) [1], which the interested reader should refer to as a framework. Within this framework, QSOD makes the real-time simulation of organizational alignment states of complex systems in Industry 4.0 possible. QSOD’s simulation as decision networks and their equivalent quantum circuits undoubtedly opens a wide field of possibilities for the study of the design of complex networked strategic organizations. As represented schematically in Figure 1a, in the mentioned work we represented the individual process owner, the node of a complex network of Industry 4.0 represented in the form of a decision graph [2], as a quantum computing unit or *qubit* [3,4]. This *qubit* is allowed to have two fundamental states, one of alignment or asymptotic stability of the key performance indicators (KPIs) defining its performance [5,6,7,8,9,10,11,12,13], represented by the state |0〉 and another of non-alignment, absence of such stability, represented by the state |1〉. In this paper we intend to expand this study, taking the simplest case of two organizational agents, a subordinate *A* reporting to another agent *B* represented in Figure 1b, that will be simulated by means of a two-*qubit* quantum circuit.

Bloch’s sphere, shown in Figure 1a, is commonly used to geometrically represent a *qubit* [14]. This is a useful and common geometric image of the quantum evolution of a single- or two-level system. On the Bloch sphere, of unitary radius, the *Z*-axis is the computational axis and its positive direction coincides with the state |0〉, and the negative with the state |1〉. A *qubit* can be represented as a point on the Bloch sphere with the help of two parameters (θ, ϕ), as expressed by Equation (Equation 1):(1)$$|\Psi \rangle =cos\left(\frac{\theta }{2}\right)|0\rangle +e^{i\phi }sin\left(\frac{\theta }{2}\right)|1\rangle$$

Our goal is to determine the probability of alignment of agent *B*, P(B=|0〉), as a function of the alignment probability of agent *A*, given by P(A=|0〉)=z, and the conditional alignment probabilities between agents *A* and *B*, given by x=P(B=|1〉|A=|0〉)∈[0,1] and y=P(B=|1〉|A=|1〉)∈[0,1]. This is achieved through the simulation of more than 500 different quantum circuit configurations in which the relative alignment probabilities x,y,z∈[0,1] vary and P(B=|0〉)=f(x,y,z) is measured. In this work we present relevant conclusions about the alignment probabilities of the higher hierarchy agent depending on the alignment state of the leadership relationship.

The rest of the work hereinafter continues as follows: first, Section 2 begins with a description of the configuration of the quantum circuit computations necessary to simulate the outlined two-*qubit* organizational design configuration. Second, Section 3 presents the case study that will simulate numerous quantum circuits, varying the mentioned parameters in order to obtain an optimal configuration of them. Third, in Section 4 we discuss the results obtained and propose an interpretation in the perspective of previous studies and of the working hypotheses. Finally, in Section 5 we discuss the findings and their implications in a broad context, and future research directions and limitations are highlighted.

## 2. QSOD Circuit—Two-Qubit Organizational Design Configuration

In this case, two *qubits* are utilized, and therefore their aggregated state can be determined utilizing the tensorial product of the individual *qubits*. The multiple *qubit* state can be expressed as a linear combination of the |0〉 and |1〉 states, then the aggregated state can be represented as in Equation (Equation 2):(2)$$|\Psi_{1}\rangle ⊗|\Psi_{2}\rangle =c_{11}c_{21}|00\rangle +c_{11}c_{22}|01\rangle +c_{12}c_{21}|10\rangle +c_{12}c_{22}|11\rangle$$
where $c_{ij}\in C^{2}$, $|\Psi_{1}\rangle =c_{11}|0\rangle +c_{12}|1\rangle$ and $|\Psi_{2}\rangle =c_{21}|0\rangle +c_{22}|1\rangle$.

Our initial hypothesis is that there is much intrinsic value for any organizational leader in Industry 4.0, to know their alignment status with the company’s strategic objectives. Moreover, not only it is important for them to know, but the company has a great interest in having its leaders aligned with its strategic objectives, since this is expected to increase its overall organizational efficiency and effectiveness. We focus on finding answers to the question of how to maximize the probability of alignment of agent *B*, P(B=|0〉), depending on agent *A*’s individual no-alignment probability, z=P(A=|1〉)∈[0,1], and the relative probability of alignment between the two agents, x=P(B=|1〉|A=|0〉)∈[0,1] and y=P(B=|1〉|A=|1〉)∈[0,1]. Mathematically speaking, we intend to find the values of (x,y,z) that maximize the function P(B=|0〉)=f(x,y,z). In other words, our challenge reduces to finding the values of x,y,z∈[0,1] that maximize Equation (Equation 3):(3)$$P\left(B=|0\rangle \right)={\left(c_{11}c_{21}\right)}^{2}+{\left(c_{11}c_{22}\right)}^{2}$$

Based on the principles of quantum circuit design that model decision networks presented in [1], the quantum circuit that models the interactions presented by Figure 1b translates into the quantum circuit Equation (4).

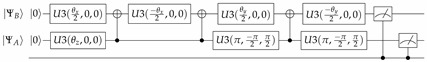
(4)

This circuit presents two *qubits*
$|\Psi_{A}\rangle$, with rotation angle $\theta_{z}$ and initial state |0〉, and $|\Psi_{B}\rangle$ with rotation angles $\theta_{x}$, $\theta_{y}$ and initial state |0〉. The respective interpretation of these rotations and the equations to calculate them are described in Table 1.

## 3. Case Study

In the case study we proceed, as announced, to simulate a total of 500 configurations of the circuit shown in Equation (4). We intend to find the values of parameters x,y,z∈[0,1] that maximize the probability of alignment of the agent *B*, P(B=|0〉). To do this, the parameters x,y∈[0,1] are varied in 10% incremental intervals in order to make a uniform mapping and create a proper display of the results. However, not all z∈[0,1] values are relevant. We are interested in values of z≥0.5, since they indicate that the alignment probability of the agent *A*, P(A=|1〉), is greater than or equal to 50%. In other words it is equal to or better than a random process. We map the values z⊂{0.5,,0.75,0.9,0.99,0.9999}, thus generating 500 simulations, each with a run of 3.5 s, giving a total computation time of 1750 s. The circuits were simulated on *qiskit* tool, a Python-based [15] quantum computing platform developed by IBM [16], and the code and results can be accessed in this Open Access Repository. We summarize the obtained results in Figure 2 by representing P(B=|0〉)=f(x,y,z) as a function of x,y,z∈[0,1].

These results can be interpreted as follows:Figure 2a describes the alignment state of agent B, P(B=|0〉), for different values of conditioned alignment probability between agents *A* and *B*, x=P(B=|1〉|A=|0〉),y=P(B=|1〉|A=|1〉)∈[0,1], being the alignment probability of agent *A*P(A=|0〉) = 1−P(A=|1〉) = 50%.Figure 2b describes the alignment state of agent B, P(B=|0〉), for different values of conditioned alignment probability between agents *A* and *B*, x=P(B=|1〉|A=|0〉),y=P(B=|1〉|A=|1〉)∈[0,1], being the alignment probability of agent *A*P(A=|0〉) = 1−P(A=|1〉) = 75%.Figure 2c describes the alignment state of agent B, P(B=|0〉), for different values of conditioned alignment probability between agents *A* and *B*, x=P(B=|1〉|A=|0〉),y=P(B=|1〉|A=|1〉)∈[0,1], being the alignment probability of agent *A*P(A=|0〉) = 1−P(A=|1〉) = 90%.Figure 2d describes the alignment state of agent B, P(B=|0〉), for different values of conditioned alignment probability between agents *A* and *B*, x=P(B=|1〉|A=|0〉),y=P(B=|1〉|A=|1〉)∈[0,1], being the alignment probability of agent *A*P(A=|0〉) = 1−P(A=|1〉) = 99%.Figure 2e describes the alignment state of agent B, P(B=|0〉), for different values of conditioned alignment probability between agents *A* and *B*, x=P(B=|1〉|A=|0〉),y=P(B=|1〉|A=|1〉)∈[0,1], being the alignment probability of agent *A*P(A=|0〉)=1−P(A=|1〉) = 99.99%.

We can intuitively observe in Figure 2 that the first partial derivatives of the two dimensional functions f(x,y)=P(B=|0〉) with changing x=P(B=|1〉|A=|0〉) and with y=P(B=|1〉|A=|1〉), given respectively by ∂f(x,y)/∂x and ∂f(x,y)/∂y. We represent the values of f(x,y)=P(B=|0〉) as a function of y=P(B=|1〉|A=|1〉) in Figure 3a. As we indicate in Figure 3b, each [0.1] interval of interval of y=P(B=|1〉|A=|1〉) contains ten values of x∈[0,1].

If we increase the granularity of the mapping of the quantum circuits in search of a fractality within their behavior, and we make a mapping of the y=P(B=|1〉|A=|1〉)∈[0,0.1] for values of z=P(A=|1〉)=0.1, then we obtain the results of Figure 4b. Similarly as in the previous diagrams, each [0.01] interval of y=P(B=|1〉|A=|1〉) contains ten values of x∈[0,1] for values of z=P(A=|1〉)=0.1.

In the following Section 4 we discuss these results in detail.

## 4. Discussion

We will now proceed to discuss the results **R** systematically. We will begin by discussing Figure 3, which describes the change in the alignment probability of the agent *B* described by the function f(x,y)=P(B=|0〉), with increasing values of the relative alignment probability of B, depending on A, given by y=P(B=|1〉|A=|1〉)∈[0,1]. Before we do so, a gentle reminder for the reader that taking into consideration Bayes’s theorem, this probability can be expressed with Equation (Equation 5):(5)$$y=P\left(B=|1\rangle |A=|1\rangle \right)=\frac{P(B=|0\rangle \cap A=|1\rangle )}{P(A=|1\rangle )}$$

This means that growing values of the relative probability of agent *B* alignment, conditioned to agent *A* being in not-alignment, are caused by growing values of the intersection P(B=|1〉∩A=|1〉). In other words, increasing values of the counter P(B=|1〉∩A=|1〉) express that the probability of the intersection of B=|0〉 and A=|1〉 is high and therefore both present similar states.

Accordingly, the following results can be enumerated:

**R1**. In general, we can say that the probability of alignment of agent *B*, f(x,y)=P(B=|0〉), oscillates consistently around the value 0.5 as a harmonic underdamped oscillator for different values of z=P(A=|1〉), which is the equilibrium state of the system. This is plausible.

**R2**. At the scale represented in Figure 3a we observe that the angular frequency of this oscillator changes for different values of y=P(B=|1〉|A=|1〉) and therefore we can separate the behavior of the function in three different regions, marked in Figure 3a, and depicted in Figure 3b in detail.

**R3**. For values of y=P(B=|1〉|A=|1〉)∈(0.2,1], the probability of alignment of agent *B*, f(x,y)=P(B=|0〉), oscillates consistently around 0.5 with a minimal amplitude for all values of z=P(A=|1〉).

**R4**. For values of y=P(B=|1〉|A=|1〉)∈(0.1,0.2], the probability of alignment of agent *B*, f(x,y)=P(B=|0〉), oscillates consistently around 0.5 with an exponential decay that consistently increases with 1−z=P(A=|0〉).

**R5**. For values of y=P(B=|1〉|A=|1〉)∈[0,0.1], the probability of alignment of agent *B*, f(x,y)=P(B=|0〉), oscillates consistently around 0.5. The oscillation presents an exponential decay for 1−z=P(A=|0〉)∈[0.5,0.9), and presents no decay for values of 1−z=P(A=|0〉)>0.9.

**R6**. However, the most striking observation of all is that if we increase the mapping of the circuits by a factor of 10, as shown in Figure 4b in which we make a mapping of the y=P(B=|1〉|A=|1〉)∈[0,0.1] with [0.01] intervals for P(A=|1〉=0.1, we observe the same behavior as with the mapping of the y=P(B=|1〉|A=|1〉)∈[0,1] with [0.1] intervals for P(A=|1〉=0.1 shown in Figure 4a. The results **R1**–**R5** are valid for both mapping intervals. The parameters (exponential decay, displacement, amplitude, and phase) of the signals represented in Figure 4 are very similar. Only the frequency is inversely proportional to the mapping interval, hence scaling the oscillation shape, and suggesting that the two graphics depict a similar process. This means that the behavior of this system is fractal. This has powerful implications, which we discuss in the conclusions.

In the following Section 5 we offer the conclusions derived from these results, we discuss the findings and their implications in a broad context, offer certain limitations of the study, and present possible next research paths to pursue.

## 5. Conclusions, Limitations and Further Steps

Quantum computing explores the processing and exchange of information as natural phenomena that respect the laws of quantum mechanics. The reason for this is that quantum computing makes use of “superposition”, which is the ability of quantum computers to simultaneously be in multiple different states. Leading an organizational effort to achieve coordinated strategic goals is a probabilistic process in which the decision makers can rarely be sure that the choice made is the right one. Managers are conditioned by the simultaneous decisions of other agents in the organization whose consequences cannot be fully anticipated a priori. The purpose of this work has been to propose an efficient quantum computing algorithm that is capable of discerning the state of alignment of the organization in the interaction of one process owner informing another and, therefore, supporting the leaders of the organizations in their decision-making process.

We can summarize the main takeaway of this study with the following statement: the case of two qubits shown by QSOD, allows us to affirm that when the strategic objective of the organizational design is to increase the alignment of a process owner, increasing the network by adding a support agent is most likely to have no effect at all on the performance of the agent being reported to. We learn this from **R1**. In fact, the probability of alignment of the original agent always oscillates around the random state.

From result **R2, R3, R4**, and **R5** we also observe that there is an exchange of energy between the original agent and the added agent, so that the alignment probability of the original agent can be positively influenced for low levels of intersection between the alignment probability of the original agent and the added agent. This could be interpreted to mean that the original agent can benefit from the presence of its new partner as long as the new partner provides process information. In other words, if the added agent is able to explain some of the variability in the value creation process that could not previously be explained by the original agent, then the asymptotic stability probability of the original agent will increase. The immediate consequence of this reading is that to add hierarchy levels to strategic design models of organizations, it is necessary to ensure the asymptotic stability of the lower agents before implementing a stable aggregation. It is important to highlight at this point that, in the context of QSOD, a hierarchy does not only describe the rather classical hierarchical relationship between agents but rather a reporting relationship. This concept is more inclusive as it includes the relationship between an agent and his/her boss, but also the relationship between an agent and his/her customer, or the relationship between an agent and a supplier. Our work helps, therefore, model interactions between organizational process owners and these interactions can potentially be scaled to any organizational context, including small and medium enterprises. This finding is very powerful for industry leaders, as well as for Strategic Organizational Design scholars because it imposes a severe constraint to ensure a sustainable and stable growth of Industry 4.0 organizations.

The implications of result **R6** are profound and reveal the essence of what some call the *fractal organization*. The interaction of two process owners reveals an energy interchange that oscillates with more or less amplitude depending on certain parameters—the conditional alignment probabilities. However, what is really striking is that, independently of the granularity in which these parameters are observed, the oscillation always follows the same pattern. Such pattern is expressed by the results **R1**–**R5** and represents the cornerstone of the bilateral interaction under study. Fractality in this QSOD context allows for a quantification of these complex dynamics and its pervasive effect offers robustness and resilience to the two-*qubit* interaction.

The main limitation of this study is that it only refers to two agents. Furthermore, the simulations of the quantum circuits have been made in a classic computer simulator. Although this undoubtedly reduces some statistical significance to the results, this fact is not relevant to our study at this time and can be neglected.

The results obtained studying the QSOD case of two *qubits* opens new interesting research questions. In order to continue offering a valuable contribution to Industry 4.0 leaders and the research community in general, the future steps we intend to take in this line of research will focus on studying the behavior of other more complex QSOD configurations.

## Figures and Tables

**Figure 1 sensors-20-06977-f001:**
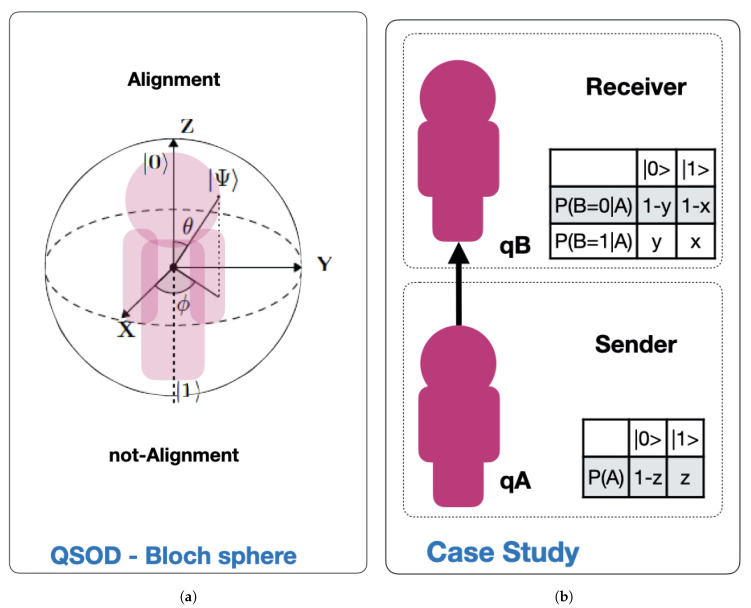
(**a**) Case study framework, (**b**) quantum simulation of strategic organizational design (QSOD)—Bloch sphere.

**Figure 2 sensors-20-06977-f002:**
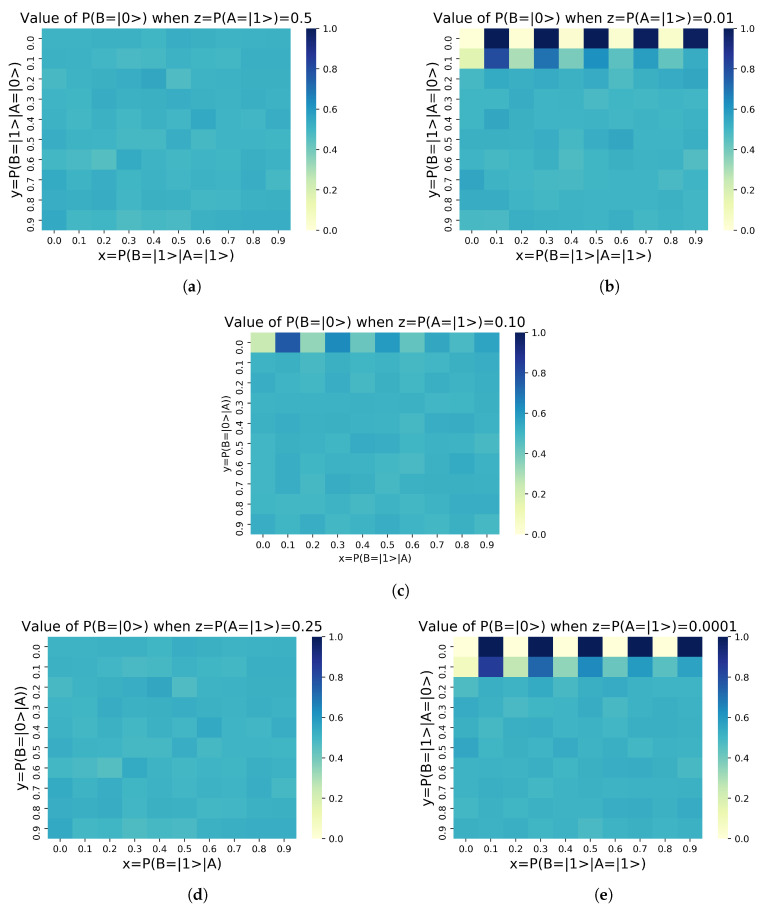
Results obtained for P(B=|0〉) for different values of the no–alignment probability of agent A,z=P(A=|1〉). (**a**) P(A=|1〉)=0.50. (**b**) P(A=|1〉)=0.25 (**c**) P(A=|1〉)=0.1 (**d**) P(A=|1〉)=0.01 (**e**) P(A=|1〉)=0.0001.

**Figure 3 sensors-20-06977-f003:**
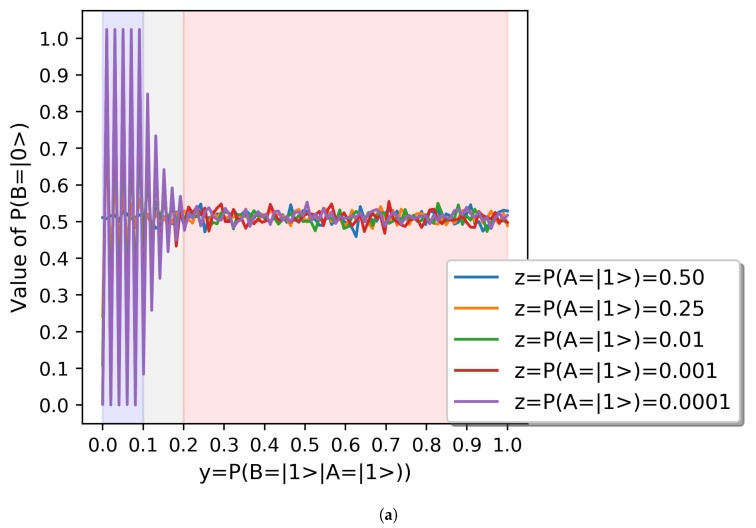
Summary of results of P(B=|0〉). (**a**) Summary of results of P(B=|0〉) as a function of y=P(B=|1〉|A=|1〉). (**b**) Detail of results of P(B=|0〉) with y=P(B=|1〉|A=|1〉)∈[0,0.3].

**Figure 4 sensors-20-06977-f004:**
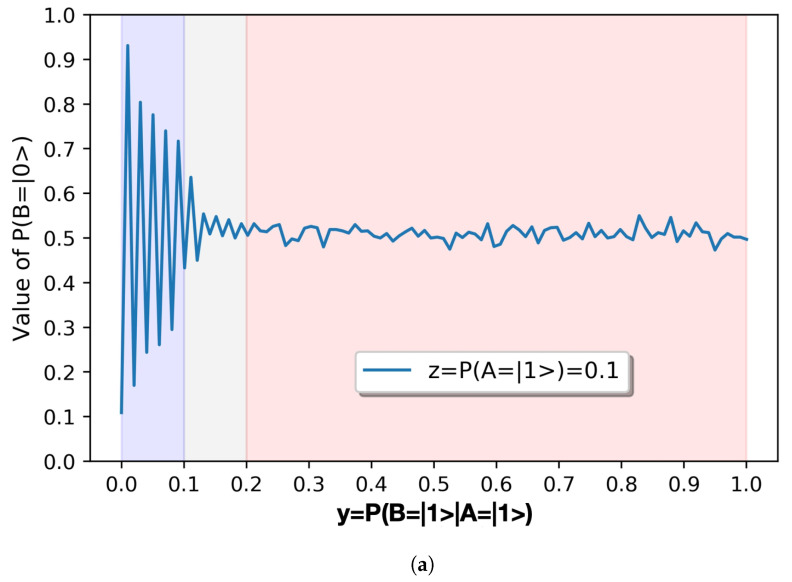
Detail of results of P(B=|0〉) for P(A=|1〉=0.1. (**a**) Detail of results of P(B=|0〉) with y=P(B=|1〉|A=|1〉)∈[0,1]. (**b**) Detail of results of P(B=|0〉) with y=P(B=|1〉|A=|1〉)∈[0,0.1].

**Table 1 sensors-20-06977-t001:** Qubit angles of rotation.

Qubit	Interpretation	Equation
$|\Psi_{A}\rangle$	The conditional probability z=P(A=|1〉) of *qubit* $|\Psi_{A}\rangle$ to be in no–alignment translates into the rotation angle $\theta_{z}$.	$\theta_{z}=2\arctan\sqrt{\frac{z}{1-z}}$
$|\Psi_{B}\rangle$	The conditional probability x=P(B=|1〉|A=|0〉) of *qubit* $|\Psi_{B}\rangle$ to be in no–alignment depending on *qubit* $|\Psi_{A}\rangle$ translates into rotation angle $\theta_{x}$.	$\theta_{x}=2\arctan\sqrt{\frac{x}{1-x}}$
	The conditional probability y=P(B=|1〉|A=|1〉) of *qubit* $|\Psi_{B}\rangle$ to be in no–alignment depending on *qubit* $|\Psi_{A}\rangle$ translates into rotation angle $\theta_{y}$.	$\theta_{y}=2\arctan\sqrt{\frac{y}{1-y}}$

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
