# Peer review of "Industry 4.0 Quantum Strategic Organizational Design Configurations. The Case of Two Qubits: One Reports to One"

_sensors, 2020, doi:10.3390/s20236977_

Round 1

Reviewer 1 Report

In this research, the authors have conducted a theoretical and analytical study on the effect of quantum strategic design on the systems communication and interfacing different parts of the the circuits. The simulation involves different probabilities and investigated 2 qubits which has open a new questions to be investigated.

The utilisation of quantum simulation allows the appotrunity to expand on the simulations of quantum circuits, this has concluded that there is an influence of the subordinate on the leader that resembles that of a harmonic underdamped oscillator around the value of 50% probability of alignment for the leader. In addition,there is a fractal behavior in this type of relationships which confirms exchange of energy between two agents that oscillates with similar amplitude with specific parameters interdependence.

The work is preliminary as there is more questions to be answered or investigated rather than concluding the work. The nature of the simulated system considered in this research is simplistic and does not exten to complex system, however, the autors have selected the option of investigatign such systems in the future work rather than doing some preliminary work to investigate more complex systems.

The manuscript is written in active voice style, please change to passive voice.
The literture is not extensive which can be expanded more to include complex systems.

The paper descripton on p2 refers to the differnet sections of the paper as chapters, it can make more sense of using sections rather than chapters. Other places, such at line 107, there is a reference ot section rather than chapter.

Some abbreviations used in the paper are not used through out the text, such as KPI. I do not see a point of making an abbreviation without beig used or cited in the text.

Line 79, there is a reference to Open Access Repositry, please provide the link.
Figure 3b, no legend, I presume it is the same as Figure 3a, but these are two different figures. You need to add another legend.

Reviewer 2 Report

As a management scholar, I have little to say on most of the paper, starting from its methods. Despite that, I had interest in reading the strategic and organizational configurations in the case of a industrial leader in relationship with a subortinated agent. What I can say is that I would appreciate a more detailed reference to the management and organizational literature when describing the implications of your results.

Reviewer 3 Report

Dear authors,

Regarding the methodology and calculations presented, I have no significant remarks.

However, I would strongly suggest improving the following parts:

-In Introduction part please provide more clearly, mainly more concretely what type of positions/relationships are you attempting to simulate, what can the reader imagine behind agents A and B (some practical examples),

-In Introduction part, please provide more clearly HOW this research design is applicable to Industry 4.0 decision networks, you should provide more studies to support your theory of linkage of your research to Industry 4.0, which is at this state not profound,

-Discussion part misses its main point from my perspective, there is no comparison to other studies´ results or conclusions with your obtained results, which is significant shortcoming of your work,

-Within the Conclusion part, again I miss any applicable, concrete, practical example of transferring your results to real company´s lifecycle,

-As you suggest adding hierarchy levels (Line 160), could you please consider taking into account also small companies with no possibility of "adding hierarchy levels",

Best regards

Round 2

Reviewer 3 Report

Dear authors,

thank you for your valuable response. At this point I would therefore suggest more obvious and more pronounced linkage of your "letter" to the "article".

Best regards,

Author Response

Dear Reviewer,

thank you very much for your valuable feedback.

We have modified the manuscript accordingly.

Best regards,

Javier